# Synovial macrophage diversity and activation of M-CSF signaling in post-traumatic osteoarthritis

**Alexander J Knights[1], Easton C Farrell[1,2], Olivia M Ellis[1,2], Michelle J Song[1,2], C Thomas Appleton[3,4,5], Tristan Maerz[1,2,6]\***

[1]Department of Orthopaedic Surgery, University of Michigan, Ann Arbor, United States; [2]Department of Biomedical Engineering, University of Michigan, Ann Arbor, United States; [3]Department of Physiology and Pharmacology, Western University, London, Canada; [4]Bone and Joint Institute, Western University, London, Canada; [5]Department of Medicine, Schulich School of Medicine and Dentistry, Western University, London, Canada; [6]Department of Internal Medicine – Division of Rheumatology, University of Michigan, Ann Arbor, United States

## eLife assessment

This study provides **useful** information by identifying the cell type (macrophages) in synovial tissues involved in the pathogenesis of post-traumatic osteoarthritis (OA) and clarifying distinct transcriptomic signatures that may be a good therapeutic target for OA. However, the analysis performed so far is **incomplete**, with a main weakness being the lack of data to confirm the authors' speculation about the underlying mechanisms.

**\*For correspondence:**
tmaerz@umich.edu

**Competing interest:** The authors declare that no competing interests exist.

**Abstract** Synovium is home to immune and stromal cell types that orchestrate inflammation following a joint injury; in particular, macrophages are central protagonists in this process. We sought to define the cellular and temporal dynamics of the synovial immune niche in a mouse model of post-traumatic osteoarthritis (PTOA), and to identify stromal-immune crosstalk mechanisms that coordinate macrophage function and phenotype. We induced PTOA in mice using a non-invasive tibial compression model of anterior cruciate ligament rupture (ACLR). Single-cell RNA-sequencing and flow cytometry were used to assess immune cell populations in healthy (Sham) and injured (7 and 28 days post-ACLR) synovium. Characterization of synovial macrophage polarization states was performed, alongside computational modeling of macrophage differentiation, as well as implicated transcriptional regulators and stromal-immune communication axes. Immune cell types are broadly represented in healthy synovium, but experience drastic expansion and speciation in PTOA, most notably in the macrophage portion. We identified several polarization states of macrophages in synovium following joint injury, underpinned by distinct transcriptomic signatures, and regulated in part by stromal-derived macrophage colony-stimulating factor signaling. The transcription factors Pu.1, Cebpα, Cebpβ, and Jun were predicted to control differentiation of systemically derived monocytes into pro-inflammatory synovial macrophages. In summary, we defined different synovial macrophage subpopulations present in healthy and injured mouse synovium. Nuanced characterization of the distinct functions, origins, and disease kinetics of macrophage subtypes in PTOA will be critical for targeting these highly versatile cells for therapeutic purposes.

## Introduction

Arthritis encompasses a host of disease manifestations in joints, characterized in part by aberrant crosstalk between stromal and immune cells, with a central unifying feature of disease being inflammation. Inflammation is driven by a diverse and highly plastic set of resident and infiltrating immune cells that can speciate into distinct functional phenotypes and engage in reciprocal crosstalk with stromal cells. While a physiological inflammatory response is integral to restoring tissue homeostasis after insults such as trauma or infection, aberrant immune cell responses underpin chronic inflammation and its associated pathological effects in many tissue contexts. For instance, early studies of diet-induced obesity in mice showed recruitment of pro-inflammatory macrophages to adipose tissue from circulating monocytes (*Lumeng et al., 2007b*; *Lumeng et al., 2007a*). In systemic sclerosis, interactions between distinct macrophage subsets and fibroblasts drive them to adopt pro-inflammatory and pro-fibrotic phenotypes, respectively, that underpin disease progression (*Fang et al., 2022*). Specifically, osteopontin produced by pathologic *SPP1+* macrophages promotes production of type I collagen by skin fibroblasts (*Gao et al., 2020*). Fibroblasts in sclerotic lesions produce high levels of TLR4 that drive persistent fibrosis (*Bhattacharyya et al., 2013*).

The synovium is a heterogenous connective tissue in joints, populated with immune cells, vascular and lymphatic endothelial cells, fibroblasts, adipocytes, and other mesenchymal-lineage stromal cells (*Lieberthal et al., 2015*). Synovium is increasingly recognized as a principal regulator of intra-articular inflammation in osteoarthritis (OA), given its highly vascularized nature and abundance of resident immune cells compared to tissues such as meniscus and cartilage. Synovial inflammation, now appreciated as a pathologic driver of OA (*Atukorala et al., 2016*; *Sanchez-Lopez et al., 2022*; *Knights et al., 2023b*), is characterized by expansion of resident synovial fibroblasts and immune cells as well as influx of systemically derived immune cells; however, we have a limited understanding of the cellular, molecular, and temporal mechanisms of these interactions, and how they differ or correspond across forms of arthritis.

The specific roles of distinct activated immune cell populations in orchestrating synovial inflammation have been well characterized in the context of rheumatoid arthritis (RA) (*Weyand and Goronzy, 2021*; *Alivernini et al., 2020*), providing valuable insights of disease pathogenesis and advancing therapeutic efficacies (*Udalova et al., 2016*). Our understanding of the pathologic immune cell landscape in post-traumatic OA (PTOA) synovium and its contribution to inflammation is only in its infancy, however.

Whether macrophage expansion is a necessary mechanism to restore homeostasis or a deleterious pathological manifestation is still controversial. Macrophage ablation in mice by clodronate liposomes yielded generally favorable OA outcomes, including reduced osteophyte formation (*Blom et al., 2007*; *Blom et al., 2004*), however pharmacogenetic macrophage depletion using the MaFIA mouse resulted in systemic inflammation, exacerbated immune influx into the joint, with no attenuation of OA severity (*Wu et al., 2017*). Indeed, synovial macrophages play critical physiological functions such as the timely clearance of apoptotic cells, which is impaired in OA (*Del Sordo et al., 2023*). In a model of inflammatory arthritis, Huang et al. showed that resolution requires the suppression of pro-inflammatory, systemically derived F4/80[hi] MHC Class II[+] macrophages and that macrophages with a resident-like phenotype promote inflammatory resolution (*Huang et al., 2021*). As such, dissecting macrophage identity, origin, and function in arthritis remains critical to better understanding disease etiology.

In this study, we employed a non-invasive anterior cruciate ligament rupture (ACLR) model of PTOA that recapitulates traumatic joint injury (*Christiansen et al., 2012*; *Maerz et al., 2015*) to dissect the dynamic immune cell environment in PTOA synovium, with a focus on the early post-injury period. Importantly, the ACLR model avoids the need for surgery and its confounding effects on synovial inflammation. We utilized single-cell RNA-sequencing (scRNA-seq) and flow cytometry to identify distinct immune cell types in healthy and PTOA synovia, with a focus on macrophages, and identified potential stromal-derived crosstalk mechanisms responsible for emergence of pro-inflammatory macrophage subtypes.

## Results

### Diverse immune cell types are present in healthy and injured synovium

Having characterized the stromal niche of synovium in PTOA and defined distinct functional subsets of fibroblasts (*Knights et al., 2023a*), we sought to understand the immune profile of healthy and injured synovium. Using our published scRNA-seq data of synovial cells from Sham, 7 days ACLR or 28 days ACLR mice (GSE211584), we computationally subset only immune cells based on their cluster identity and verified their positivity for *Ptprc* expression (encoding CD45) and negativity for *Pdgfra* expression (encoding PDGFRα) (*Figure 1—figure supplement 1A and B*). In the absence of stromal and vascular cells that would otherwise confound variance, re-clustering was performed on the immune cells, revealing a diverse profile of innate and adaptive cell types, including monocytes and macrophages, dendritic cells (DCs), T cells, B cells, mast cells, granulocytes, innate lymphoid cells (ILCs), and natural killer (NK) cells (*Figure 1A*, *Figure 1—figure supplement 1C*). Abundance of synovial immune cells greatly increased acutely after injury (7 days ACLR), with a degree of resolution occurring by 28 days ACLR (*Figure 1B and C*). Macrophages, unsurprisingly, predominated both numerically and proportionally in healthy and injured synovium, with various subsets of T cells and DCs comprising most of the remaining cells (*Figure 1C*, *Figure 1—figure supplement 1D–F*). To assess whether the immune cell types observed in scRNA-seq based on their transcriptomic signatures were detectable at the functional protein level using surface markers, we employed multi-color flow cytometry. Indeed, all major groups of immune cells were identified within the CD45+ pool of synovial cells, based on their surface marker and light scatter profiles (*Figure 1D and E*).

### Integration of joint immune cells from mouse scRNA-seq datasets

Given the recent proliferation of studies seeking to describe the immune landscape in murine arthritis models, there is a strong rationale for harmonizing this publicly available data to characterize similarities and differences. We obtained four published scRNA-seq datasets for integration and further analysis of the immune cell niche: *Culemann et al., 2019* (GSE134420; sorted CD45+ CD11b+ Ly6G- cells from hindpaw synovia of mice subjected to the K/BxN serum transfer arthritis model of inflammatory arthritis/RA); *Sebastian et al., 2022* (GSE200843; sorted CD45+ cells from whole knee joints of mice subjected to the ACLR model of PTOA); our data, from *Knights et al., 2023a* (GSE211584; knee synovium from mice subjected to the ACLR model of PTOA); and *Muench et al., 2022* (GSE184609; sorted live cells from hindpaw synovia of mice subjected to the GPI-induced model of inflammatory arthritis/RA).

Datasets were obtained from the NCBI Gene Expression Omnibus then recapitulated in accordance with parameters provided in the Materials and methods sections of each paper, including computational removal of any non-immune cell clusters (*Figure 2—figure supplements 1–3*). The resulting immune cell-only objects were integrated using reciprocal PCA and assigned identities based on known marker genes and alignment to published transcriptomic signatures of immune cell types using the Cluster Identity PRedictor (CIPR) (*Ekiz et al., 2020*; *Figure 2A and B*, *Figure 2—figure supplement 4A*). Monocytes, macrophages, DCs, neutrophils, B cells, T and NK cells, mast cells, and proliferating immune cells were all detected, and the same, or very similar, cell types clustered closely together across datasets following integration (*Figure 2A–D*). Likely owing to the disparate models (inflammatory arthritis/RA or PTOA), anatomical locations (knee synovium, whole knee joint, or hindpaw synovium), and experimental approaches (e.g. sorting only monocytes and macrophages), certain cell types predominated in some datasets but were largely absent in others. This was most strikingly evident with the neutrophil clusters (Neut-1, -2, and -3) present primarily in the Sebastian and Muench datasets, the latter of which modeled RA in which neutrophils are known to play a larger role (*Khandpur et al., 2013*). One explanation for high neutrophil abundance in these datasets may be bone marrow contamination, given that these studies digested the whole knee joint (Sebastian) or the whole hindpaw (Muench). We detected a high abundance of neutrophils in digested whole knee joints by flow cytometry – approximately half of all CD45+ cells (*Figure 2—figure supplement 5A*). This number was not reduced when mice were systemically perfused to clear the vasculature, suggesting that neutrophils in the joint digest are not derived from local blood vessels. Comparison of the gene signatures of neutrophil clusters 1, 2, and 3 to publicly available neutrophil datasets support that these cells, particularly Neut-1, are more likely bone marrow-derived than true intra-articular tissue neutrophils (*Figure 2—figure supplement 5B and C*, *Supplementary file 2*).

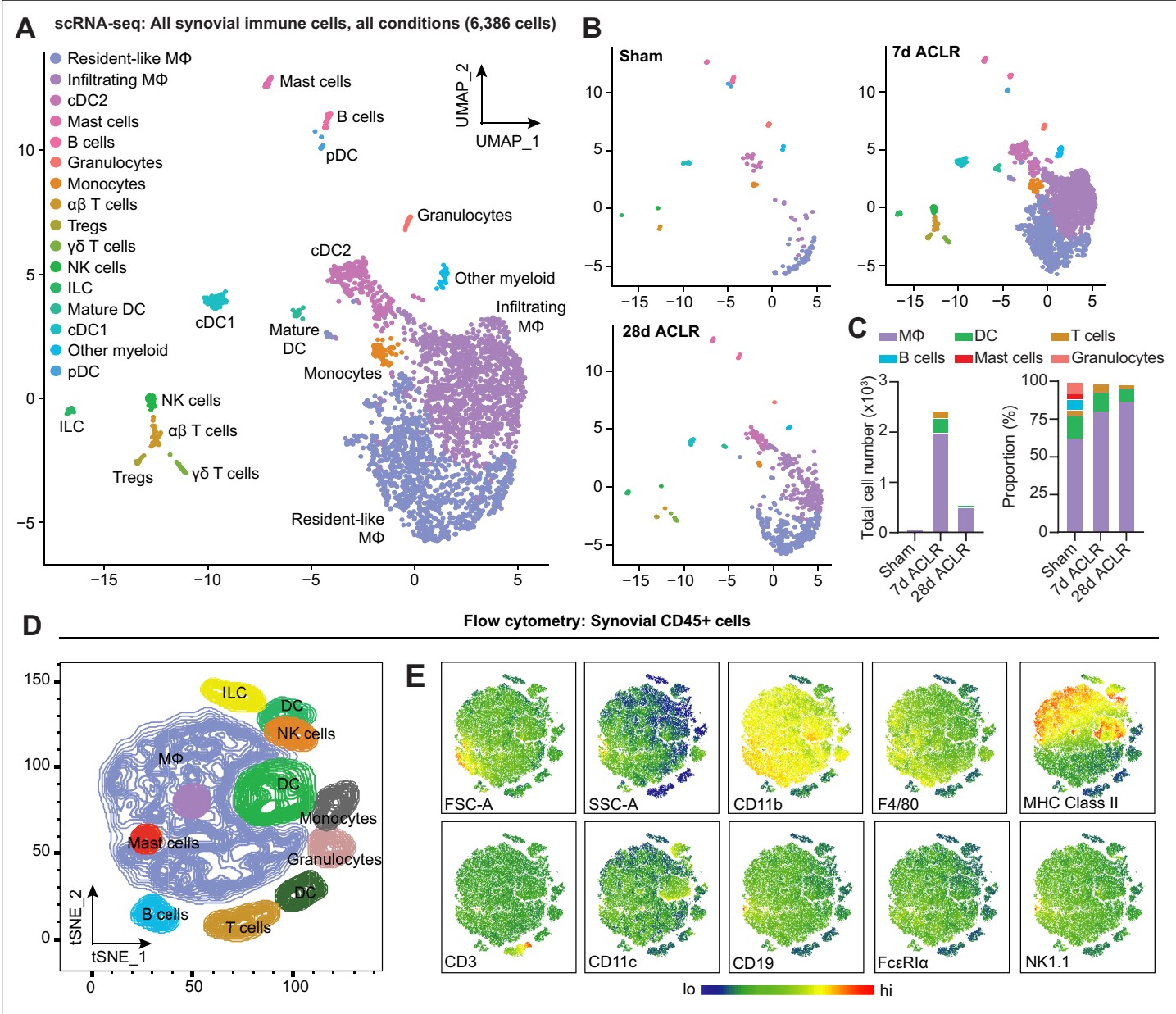

**Figure 1.** Diverse immune cell types are present in healthy and injured synovium. (**A**) Uniform manifold approximation and projection (UMAP) plot of all immune cells by single-cell RNA sequencing (scRNA-seq) of synovium from mice subjected to Sham, 7 days anterior cruciate ligament rupture (ACLR) and 28 days ACLR, or (**B**) split by condition. (**C**) Breakdown of immune cell types by total abundance (left) or proportion (right) in Sham, 7 days ACLR or 28 days ACLR synovium. (**D**) t-Distributed stochastic neighbor embedding (t-SNE) plot of immune cell types by flow cytometry of CD45+ synovial cells from mice subjected to Sham (left and right synovia from n=4 mice) and 7 days ACLR (right synovia from n=4 mice). (**E**) t-SNE heatmaps of scatter and surface marker parameters used to define synovial immune cell identities by flow cytometry. Tregs: regulatory T cells; NK cells: natural killer cells; ILC: innate lymphoid cells; DC: dendritic cells; cDC: conventional DCs; MΦ: macrophages; FSC-A: forward scatter area; SSC-A: side scatter area.

The online version of this article includes the following figure supplement(s) for figure 1:

**Figure supplement 1.** Computational subsetting of immune cells from whole synovium single-cell RNA sequencing (scRNA-seq).

Across datasets, myeloid cells (monocytes, macrophages, granulocytes, DCs) were the predominant cell group, with lymphocytes (T and B cells) comprising less than 20% of total immune cells, and mast cells were very rare (*Figure 2C*, *Figure 2—figure supplement 4B*). To assess broad signaling patterns of immune cell types across datasets and disease states, we used CellChat (*Jin et al., 2021*). River plots of outgoing communication signals from immune cell clusters showed distinct broad signaling patterns for most immune cell types across datasets, with the notable exception of DCs and

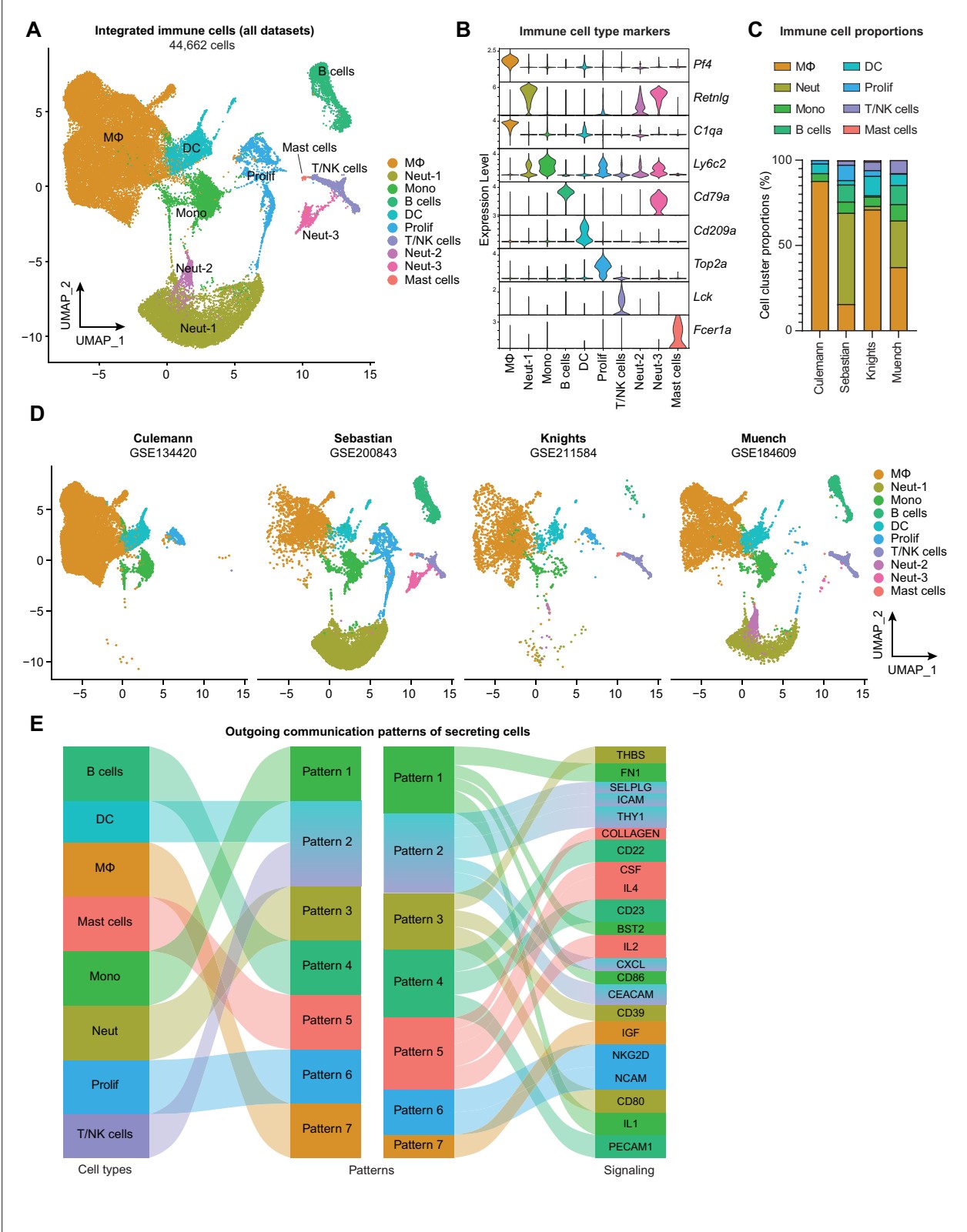

**Figure 2.** Integration of joint immune cells from mouse arthritis single-cell RNA sequencing (scRNA-seq) datasets. (**A**) Integrated uniform manifold approximation and projection (UMAP) plot of all immune cells from GSE134420 (Culemann), GSE200843 (Sebastian), GSE211584 (Knights), and GSE184609 (Muench). (**B**) Violin plots showing gene markers for each cell cluster. (**C**) Proportional breakdown of each major immune cell type in each

*Figure 2 continued on next page*

*Figure 2 continued*

dataset. (**D**) UMAP plots showing immune cell clusters for each dataset. (**E**) CellChat outgoing communication patterns for each major immune cell group. Neut: neutrophils; Mono: monocytes; Prolif: proliferating cells.

The online version of this article includes the following figure supplement(s) for figure 2:

**Figure supplement 1.** Recapitulation of published single-cell RNA sequencing (scRNA-seq) datasets: Culemann.

**Figure supplement 2.** Recapitulation of published single-cell RNA sequencing (scRNA-seq) datasets: Sebastian.

**Figure supplement 3.** Recapitulation of published single-cell RNA sequencing (scRNA-seq) datasets: Muench.

**Figure supplement 4.** Integration of immune single-cell RNA sequencing (scRNA-seq) datasets.

**Figure supplement 5.** Assessment of neutrophils in the whole joint.

T/NK cells which were all predicted to participate strongly in signaling via integrins, selectins, and chemokine ligands (*Figure 2E*, *Figure 2—figure supplement 4C and D*). Mast cells were enriched for sending out CSF and interleukin-4 (IL-4) signals, while monocytes and macrophages were enriched for IL-1 and IGF signaling.

## Comparison of macrophages between disease states

Much attention has been devoted to the similarities and differences between OA and RA (*Zhang et al., 2022*; *Ravi et al., 2012*; *Huang et al., 2019*; *Pap and Korb-Pap, 2015*), however less focus has been directed at how intra-articular macrophages compare between the two diseases. From the integrated object containing immune cell clusters from all four datasets, we subset only monocytes and macrophages (*Figure 3A*). The resulting three clusters corresponded to monocytes (*Plac8* and *Ly6c2*), macrophages (*C1qa* and *Adgre1*), and osteoclast-like cells (*Ctsk* and *Acp5*) (*Figure 3B*). Macrophages represented a higher proportion of cells in the inflammatory/RA datasets (Culemann and Muench), while monocytes were more proportionally abundant in the PTOA datasets (Sebastian and Knights) (*Figure 3C and D*, *Figure 3—figure supplement 1A*). Osteoclasts represented <5% of cells, and macrophages were vastly more abundant than monocytes or osteoclast-like cells in both PTOA and RA models.

CellChat analysis of outgoing signaling patterns for each cluster in PTOA and RA highlighted monocytes as dominant drivers of signaling with a high degree of commonality between diseases (*Figure 3—figure supplement 2A-D*). Macrophages and osteoclasts, on the other hand, exhibited more disparate outgoing signaling programs between PTOA and RA. To better understand transcriptomic similarities and differences between macrophages from PTOA and RA synovial joints, we next performed two differential gene expression analyses: (i) PTOA vs healthy control macrophages (Sebastian and Knights) and (ii) RA vs healthy control macrophages (Culemann and Muench). 85 differentially expressed genes (DEGs) were found exclusively between PTOA and control macrophages, 89 DEGs were found exclusively in the RA vs control comparison, and 21 DEGs overlapped between both analyses (*Figure 3E*, *Supplementary files 3 and 4*). Thus, to determine perturbed pathways unique to macrophages in each disease state separately, we used exclusive DEGs to perform pathway analysis using Gene Ontology (*Supplementary file 5*). In PTOA macrophages, compared to controls, collagen metabolism, myeloid differentiation regulation, neutrophil chemotaxis, and glial cell apoptosis pathways were all enriched (*Figure 3F*). In RA macrophages on the other hand, response to corticotropin-releasing hormone (known to be pro-inflammatory and enriched in RA synovial fluid; *Malysheva et al., 2012*), Th1 differentiation, and cellular response to mercury were all perturbed (*Figure 3G*). Analysis of enriched pathways common between RA and PTOA macrophages yielded unsurprising results, including cytokine activity, white blood cell chemotaxis and migration, and signaling receptor activity (*Figure 3H*).

## Synovial macrophage subsets and trajectories in PTOA

In *Figure 1* we identified macrophages and monocytes as the most abundant immune cell group in both healthy and PTOA synovium. To eliminate variation from other cell types and to more sensitively describe transcriptional variance among macrophages, we computationally isolated monocytes and macrophages from all immune cells in our scRNA-seq dataset (*Figure 4A*, *Figure 4—figure supplement 1A*). Five clusters emerged, expressing classical monocyte and macrophage markers *Itgam* (encoding CD11b), *Adgre1* (encoding F4/80), *Csf1r* (encoding the M-CSF receptor), and *Cx3cr1*

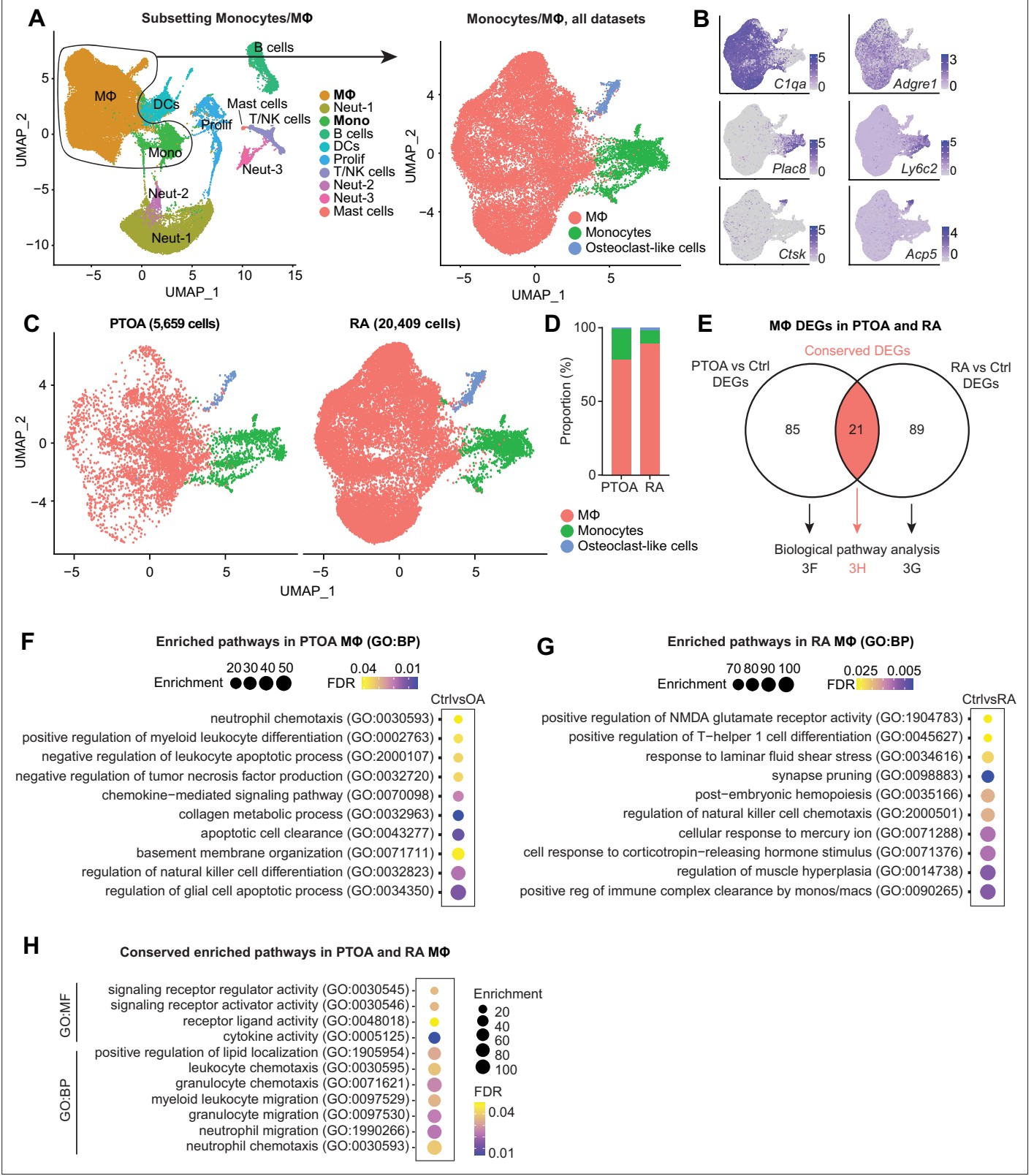

**Figure 3.** Comparison of macrophages between arthritis disease states. (**A**) Monocytes and MΦ from all post-traumatic osteoarthritis (PTOA) and rheumatoid arthritis (RA) immune cell datasets were computationally isolated, resulting in three clusters: MΦ, monocytes and osteoclast-like cells. (**B**) Feature plots showing expression levels of highly enriched genes unique to each cluster. (**C**) Side-by-side uniform manifold approximation and projection (UMAP) plot of monocytes and MΦ from PTOA and RA datasets and (**D**) the proportion of each cluster within each disease state.

*Figure 3 continued on next page*

*Figure 3 continued*

(**E**) Differential gene expression analyses were performed on the MΦ cluster specifically. Comparison groups were PTOA vs control (Ctrl) MΦ and RA vs control (Ctrl) MΦ, with overlapping and non-overlapping differentially expressed genes (DEGs) for each comparison shown in a Venn diagram. Also see *Supplementary files 3 and 4* . (**F–G**) Enriched biological pathways in PTOA (**F**) and RA (**G**) MΦ when compared to their respective control MΦ. (**H**) Conserved enriched biological pathways between PTOA and RA MΦ, derived from common DEGs with the same directionality between both separate comparisons. Also see *Supplementary file 5* . For all pathway analyses, statistical overrepresentation tests were performed with Fisher's exact testing and calculation of false discovery rate (FDR). GO:BP: Gene Ontology Biological Pathways; GO:MF: Gene Ontology Molecular Function.

The online version of this article includes the following figure supplement(s) for figure 3:

**Figure supplement 1.** Macrophages in osteoarthritis (OA) and rheumatoid arthritis (RA).

**Figure supplement 2.** Outgoing signaling patterns from macrophages in post-traumatic osteoarthritis (PTOA) and rheumatoid arthritis (RA).

(*Figure 4A*, *Figure 4—figure supplement 1B*). Sham synovium was predominated by macrophages expressing *Timd4*, *Lyve1*, and *Folr2* which we have termed basal resident macrophages, given their resemblance to homeostatic macrophages residing in various tissues with the same gene expression signature (*Chakarov et al., 2019*; *Dick et al., 2022*; *Figure 4B and C*). After injury, there was a profound increase in cellular abundance and heterogeneity, but with the notable loss of cells characterized by the basal resident gene signature. Trajectory modeling using Monocle3 indicated that basal resident macrophages polarized into distinct phenotypes in PTOA, characterized by expression of *Mrc1*, *Ly6e*, and *Gas6* (resident-like A) or *Cd9*, *Spp1*, and *Trem2* (resident-like B) (*Figure 4C and D*).

The top DEGs between resident-like macrophage clusters were analyzed to better understand their potential functional differences (*Figure 4E*, *Supplementary file 6*). Compared to basal resident macrophages, the resident-like A cluster was enriched for functions pertaining to complement activation and cell migration (*Figure 4F*). Resident-like B cells had enriched CXCL12-CXCR4 signaling and production of macrophage inflammatory protein, compared to the basal resident cells. On the other hand, compared to the two polarized resident-like macrophage subsets, basal resident cells had functional enrichment of homeostatic biological pathways including phagocytosis, endocytosis, and apoptosis. The resident-like A subset was enriched for antigen presentation functions compared to resident-like B.

Monocytes expressing *Ly6c2*, *Plac8*, and *Chil3* were also present, particularly after injury, and trajectory analysis suggested that they give rise to a pro-inflammatory cluster expressing *Ccr2* and genes encoding MHC Class II, which we have termed infiltrating macrophages. Genes historically used to mark 'M1' or 'M2' macrophages, including *Tnf* (encoding TNF-α) and *Il10* (encoding IL-10), did not segregate neatly into defined clusters (*Figure 4—figure supplement 1C and D*), emphasizing the need to supersede these outdated terms with more nuanced characterizations based on their in vivo gene expression and functional profiles.

Together these results suggest that functional polarization states of macrophages arise from a resident pool of cells during PTOA, and that a pro-inflammatory infiltrating macrophage population derives from circulating monocytes. Importantly, however, in vivo approaches such as lineage tracing will be required to assess the veracity of in silico trajectory modeling.

## Stromal-immune crosstalk via M-CSF signaling

Stromal and immune cells participate in autocrine and paracrine signaling within synovium, and we sought to define key signaling axes that regulate the synovial immune cell niche. Using CellChat ligand-receptor interactions for all synovial cell types defined as stromal or immune, we implemented a novel approach to derive crosstalk communication scores for multi-directional and uni-directional stromal-immune signaling (*Figure 5A*). The M-CSF pathway, involving the ligands M-CSF (macrophage colony-stimulating factor) and IL-34, and the M-CSF receptor, was highly induced after injury, but primarily in the stromal-to-immune direction (*Figure 5B*, *Figure 5—figure supplement 1A-F*), implying that stromal-derived ligands regulate synovial immune cells expressing the M-CSF receptor – namely, macrophages. M-CSF and its receptor are well recognized for their role in monocyte and macrophage function (*Lin et al., 2019*; *Guilliams et al., 2020*), and more recently, IL-34 was identified as an alternative M-CSF ligand (*Ma et al., 2012*; *Lin et al., 2008*). Bulk RNA-seq of synovium showed robust induction of *Csf1* (encoding M-CSF), *Il34* (encoding IL-34), and *Csf1r* (encoding the M-CSF receptor) after joint injury (*Figure 5C*); thus, we sought to determine the cell types orchestrating this signaling axis.

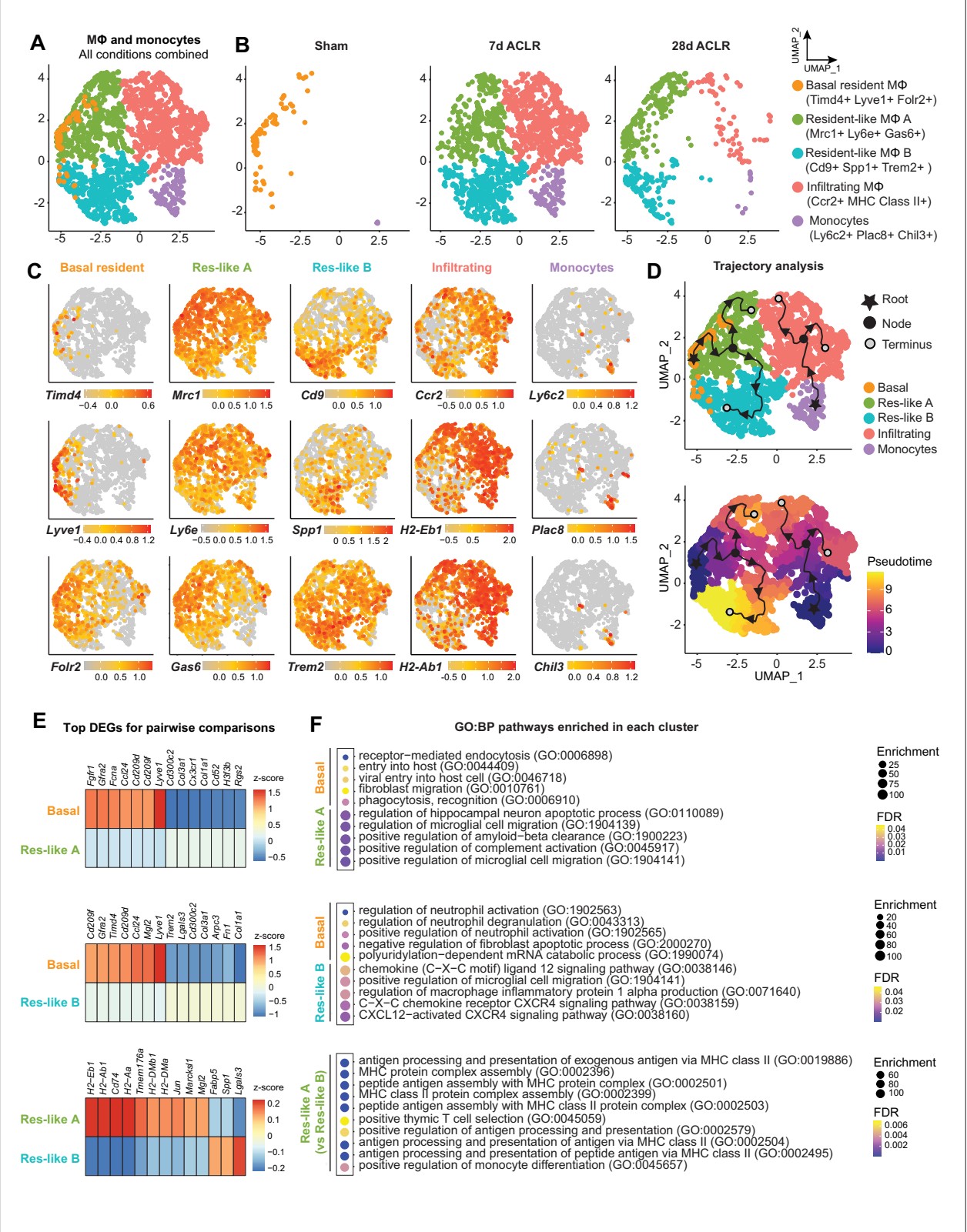

**Figure 4.** Synovial macrophage subsets and trajectories in post-traumatic osteoarthritis (PTOA). (**A**) Uniform manifold approximation and projection (UMAP) plot of monocytes and MΦ from synovium of Sham, 7 days anterior cruciate ligament rupture (ACLR) and 28 days ACLR mice, or split by condition (**B**). Cluster naming and top gene markers are given on the right. (**C**) Gene feature plots showing expression of key marker genes for each subset. (**D**) Pseudotime trajectories overlaid onto monocyte and MΦ subsets showing directionality (arrowheads), starting points (roots, stars), branching

*Figure 4 continued on next page*

*Figure 4 continued*

points (nodes, black circles), and endpoints (termini, gray circles with black outline). Partitions are shown as disconnected (separate) trajectory trails. Colored cell clusters are shown in the top plot and pseudotime scale is shown in the bottom plot. (**E**) Heatmaps of top differentially expressed genes (DEGs) in pairwise comparisons for basal resident MΦ, resident-like MΦ A, and resident-like MΦ B clusters (padj<0.05). (**F**) Enriched biological pathways in basal resident MΦ, resident-like MΦ A, and resident-like MΦ B clusters, derived from statistical overrepresentation tests of DEGs from corresponding pairwise comparisons in (**E**). Also see *Supplementary file 6*. Fisher's exact testing was performed and false discovery rate (FDR) was calculated. GO:BP: Gene Ontology Biological Pathways.

The online version of this article includes the following figure supplement(s) for figure 4:

**Figure supplement 1.** Synovial macrophage subsets and trajectories.

CellChat hierarchy plots, which infer directionality and degree of signaling between each cell type based on ligand-receptor interactions of a given pathway, were generated for all synovial cells from Sham, 7 days ACLR, and 28 days ACLR. In Sham synovium, lining and sublining fibroblasts, as well as pericytes, were the primary producers of M-CSF and/or IL-34 ligand, with macrophages and DCs being the primary signal-receiving cell types (*Figure 5D*). After injury, alongside the increased cell type diversity, there were more M-CSF pathway interactions (number of connections) with higher probabilities (thickness of connections), which, again, originated from fibroblasts and pericytes and signaled toward myeloid cells. A small population of mast cells also sent outgoing signals at 7 days ACLR only. No immune-to-stromal signaling via the M-CSF pathway was predicted (*Figure 5—figure supplement 2A-C*). Expression patterns of M-CSF pathway genes in synovial cell types confirmed fibroblasts as the major source of *Csf1* ligand, with no expression seen in immune cells (except for mast cells); pericytes were the primary source of *Il34* ligand; and *Csf1r* expression was confined to the immune cell compartment (*Figure 5E–G*). These results shed light on the cellular participants, temporal dynamics, and regulation of M-CSF signaling in synovium, which is among the most activated stromal-immune crosstalk axes following joint injury.

## Transcriptional control of monocyte differentiation in synovium

Canonical monocyte-to-macrophage differentiation is well characterized, however the regulation of this phenomenon in synovium remains largely undescribed. Thus, we next sought to model the differentiation mechanism of blood-derived monocytes that enter synovium and give rise to infiltrating macrophages. The trajectory from *Figure 4D* was isolated to only include monocytes to infiltrating macrophages, eliminating unrelated variance stemming from resident macrophage clusters (*Figure 6A and B*). Modules of genes co-regulated across pseudotime were generated using Monocle3 (*Figure 6C*, *Supplementary file 7*), which represent the various transcriptional programs activated during monocyte-to-macrophage maturation, and the modules most highly enriched at the trajectory terminus were further analyzed (modules 1, 4, and 6).

Given that genes in these modules are induced in the terminal stage of monocyte-to-macrophage differentiation in synovium, we screened for which transcription factors (TFs) control this fate. Genes from modules 1, 4, and 6 were analyzed by RcisTarget, which screens gene promoters for overrepresented binding motifs, allowing us to infer TFs that may putatively control monocyte-to-macrophage differentiation in synovium. After filtering out putative factors showing little or no expression, the top-ranked candidates for regulating differentiation were Pu.1 (Spi1), Cebpα, Cebpβ, and Jun (*Figure 6D*), all of which have reported roles in controlling macrophage fate and function (*Zhang et al., 1994*; *Hannemann et al., 2017*; *Yeamans et al., 2007*; *Ruffell et al., 2009*). The gene regulatory network for these TFs (*Figure 6E*) showed distinct (e.g. *Il10*, *Rel*, *Tlr7*, *Egr2*) and shared (e.g. *Vegfa*, *Irf2*, *Il6ra*, *Nlrp3*) target genes, with Pu.1 controlling expression of *Csf1r* as has been described in other settings (*Zhang et al., 1994*; *Aikawa et al., 2010*). Expression of the genes encoding Pu.1, Cebpα, Cebpβ, and Jun was not necessarily induced over pseudotime, however their gene targets (derived from modules 1, 4, and 6), such as *Il10*, *Csf1r*, and *Tlr7*, showed increasing expression with pseudotime (*Figure 6F*, *Figure 6—figure supplement 1A*). Together, these results point toward four key TFs that control the differentiation of blood-derived monocytes into pro-inflammatory synovial macrophages and identified the genes they regulate during this process, while corroborating our understanding of canonical monocyte differentiation and how it is controlled.

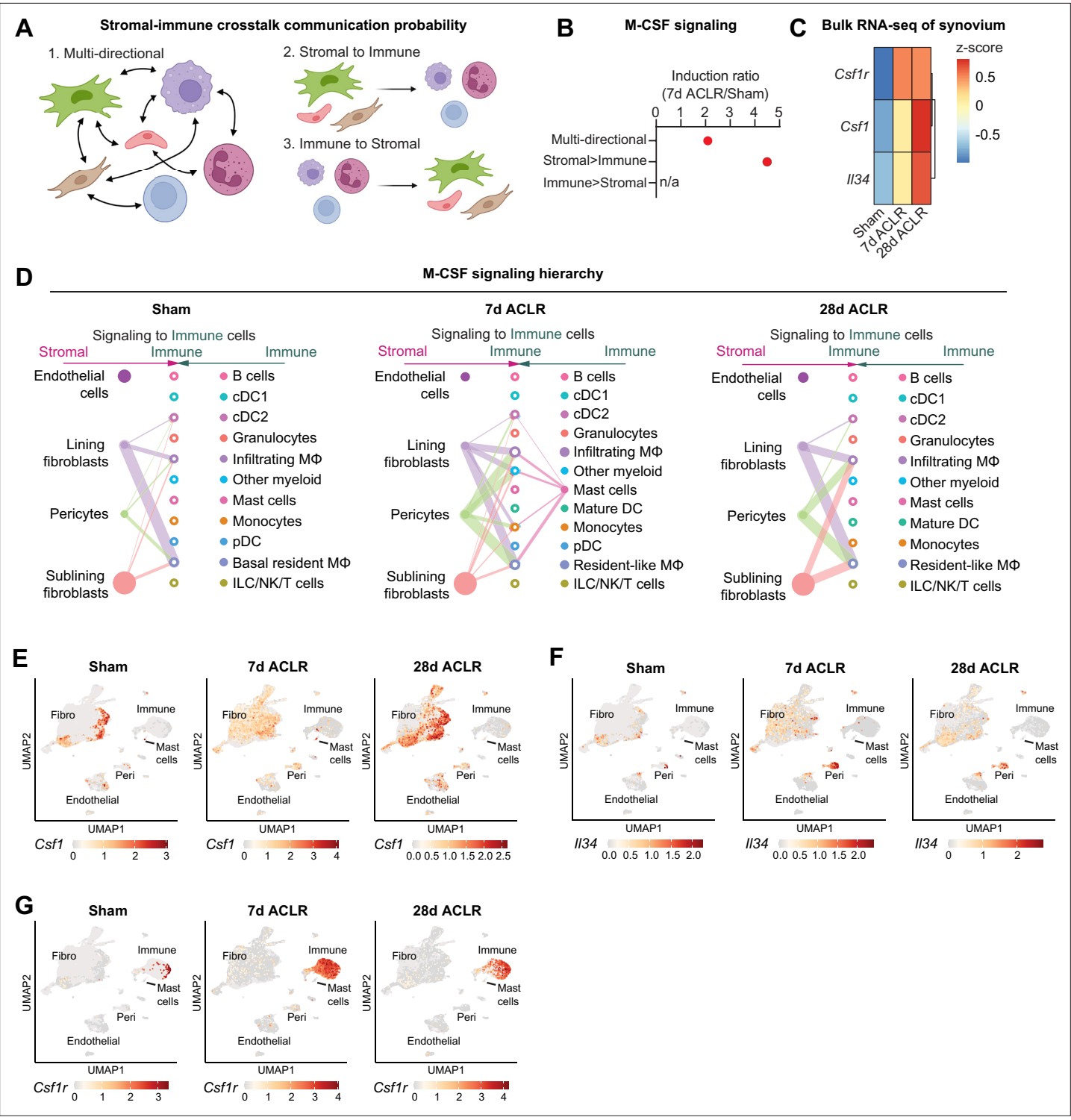

**Figure 5.** Stromal-immune crosstalk via M-CSF signaling. (**A**) Cartoon depicting multi-directional and uni-directional crosstalk between stromal and immune cells in synovium, used for calculation of crosstalk communication probability scores. (**B**) Induction ratio for multi-directional and uni-directional M-CSF signaling derived from crosstalk communication probability scores (7 days anterior cruciate ligament rupture [ACLR]/Sham). (**C**) Expression of M-CSF ligands *Csf1* and *Il34*, and the M-CSF receptor *Csf1r*, from bulk RNA-seq of Sham, 7 days ACLR, or 28 days ACLR synovium (n=5–6 male and n=5–6 female synovia per condition) (dataset available at NCBI GEO Accession Number GSE271903). (**D**) CellChat hierarchy plots for the M-CSF signaling pathway in Sham, 7 days ACLR and 28 days ACLR. Circles with fill represent cells sending signals, circles without fill represent cells receiving signals, in each condition. Line thickness corresponds to strength of communication. (**E–G**) Feature plots showing expression of *Csf1* (**E**), *Il34* (**F**), and *Csf1r* (**G**) in all synovial cells, split by condition (Sham, 7 days ACLR, or 28 days ACLR). n/a: not applicable; Fibro: fibroblasts; Peri: pericytes.

*Figure 5 continued on next page*

*Figure 5 continued*

The online version of this article includes the following figure supplement(s) for figure 5:

**Figure supplement 1.** Stromal-immune crosstalk communication probability ratios in Sham and 7 days anterior cruciate ligament rupture (ACLR).

**Figure supplement 2.** M-CSF signaling hierarchy to stromal cells.

## Discussion

Chronic synovitis is recognized to drive various pathological processes associated with arthritis, including pain and tissue damage, but the early cellular and molecular events following joint injury are not well described. We recently demonstrated that synovial fibroblasts expand rapidly and phenotypically diverge into distinct functional subsets in PTOA, alongside pathological activation of canonical Wnt signaling (*Knights et al., 2023a*). The present work demonstrates the injury-induced immune cell dynamics concurrent with the activation of synovial fibroblasts, and our flow cytometric and transcriptomic analyses demonstrate that while most immune cells are rare in healthy synovium, joint injury causes massive diversification and influx of immune subsets. Macrophages exhibited the greatest expansion, and our single-cell transcriptomic analyses identified a resident macrophage population that polarizes into two phenotypes following injury, in addition to an infiltrating macrophage population derived from monocytes. Synovial stromal cells were highly active in mediating immune cell infiltration and expansion, with unbiased modeling of stromal-immune crosstalk identifying that M-CSF signaling is among the most perturbed ligand-receptor axes after injury. The TFs Jun, Cebpα, Cebpβ, and Pu.1 were identified as putative mediators of differentiation from monocytes to synovial macrophages.

Among the most pressing questions in the field of PTOA research is which immune cells drive disease vs which promote resolution following joint injury. Rigorously describing immune cell dynamics and the relative contribution of specific cellular subsets to disease processes such as pain, osteophyte formation, and cartilage damage will facilitate the development of targeted, disease-modifying therapies. Consistent with prior studies (*Sebastian et al., 2022*; *Haubruck et al., 2020*; *Furman et al., 2021*), our transcriptomic profiling demonstrated that healthy synovia contained all major immune cell types, including rare subsets such as mast cells and T cells, and this was corroborated by flow cytometry. As we did not perfuse mice prior to tissue collection, some of these may be blood-derived, but mature immune cells such as mast cells, T cells, B cells, and DCs are unlikely to be present in healthy synovial tissue vasculature in sufficient numbers to generate this result. Robust lineage tracing studies are lacking to describe the origin, maintenance, and replenishment of these immune cells, which will require more specific tools given that recent scRNA-seq studies have revealed that many of the existing genetic reporter systems are insufficient to distinguish between resident vs systemically derived cells, or exhibit too much overlap in expression among different immune subsets (e.g. *LysM-cre*, *Cx3cr1-cre*, *Ccr2-cre*).

Macrophages were the most numerous immune cell type in healthy and injured synovium and exhibited the greatest expansion following injury. In healthy tissue, we identified a resident synovial macrophage population with a *Timd4+ Lyve1+ Folr2+* (TLF) signature that resembled homeostatic tissue-resident macrophages described in other tissue contexts (*Chakarov et al., 2019*; *Dick et al., 2022*). Trajectory analysis predicted that these resident macrophages could expand into two injury-induced resident-like subsets: one marked by *Mrc1* (encoding CD206) and *Gas6*, and another marked by *Cd9*, *Spp1*, and *Trem2* expression. Several recent studies have corroborated the existence of a pro-fibrotic macrophage population that emerges in different disease contexts (*Gao et al., 2020*; *Alivernini et al., 2020*; *Zhang et al., 2019*; *Hoeft et al., 2023*; *Hill et al., 2018*; *Jaitin et al., 2019*; *Fujii et al., 2022*; *Ramachandran et al., 2019*), expressing *Spp1* (encoding osteopontin), *Cd9*, and *Trem2*, strongly suggesting that the resident-like B macrophage population we identified in PTOA synovium is functionally analogous, which is significant in light of the strong fibrotic phenotype of PTOA synovium. Our trajectory modeling pointed toward these pro-fibrotic macrophages arising from synovium-resident TLF+ macrophages after injury. In liver cirrhosis, however, trajectory analysis suggested a monocytic origin for *Spp1*+ macrophages. Thus, robust in vivo fate mapping studies are critical to elucidate their true origins. The resident-like A cluster that emerged after injury expressed *Mrc1* (CD206), historically associated with an anti-inflammatory and reparative phenotype, and which has been shown to mediate collagen turnover (*Madsen et al., 2013*; *Wang et al., 2020*). In addition,

their potentially beneficial role in regulating extracellular matrix, the resident-like A population also expressed *Gas6*, which is required for efficient efferocytosis of apoptotic cells by synovial macrophages (**Yao et al., 2023**) – an important aspect of inflammatory resolution that goes awry in OA (**Del Sordo et al., 2023**). It is noteworthy that conventional 'M1' and 'M2' markers, such as *Il1b*, *Tnf*, *Arg1*, and *Il10*, revealed no appreciable separation across our clusters, placing further impetus to move beyond the simplistic M1-M2 spectrum of macrophages. Mechanistic and in vivo studies are now warranted to describe the precise pathophysiological functions of these subsets in OA, their ontogenies, and their spatial distribution in the various synovial niches.

We further identified a clear monocyte-to-macrophage trajectory, and these infiltrating macrophages highly expressed the pro-inflammatory genes encoding for CCR2, IL-1β, TNF-α, MHC Class II, among others. Huang et al. recently demonstrated in murine inflammatory arthritis that synovial inflammatory resolution is achieved, in part, by suppressing the in situ differentiation of infiltrating monocytes to an F4/80$^{hi}$ MHC Class II$^+$ phenotype, which would otherwise promote chronic inflammation (**Huang et al., 2021**). Thus, we conclude that the monocyte-derived macrophages we identified to emerge in synovium following joint injury are pro-inflammatory and analogous to those described by Huang et al., and developing an understanding of the molecular mechanisms that promote their recruitment and differentiation following joint injury is critical. We therefore undertook further focused analysis of the transcriptional trajectory spanning monocytes to infiltrating macrophages, and we identified Jun, Cebpα, Cebpβ, and Pu.1 as TFs potentially regulating the gene programs of monocyte-to-macrophage maturation in OA synovium. These have been previously attributed to regulate macrophage maturation in other disease contexts such as RA (**Tu et al., 2023**; **Hannemann et al., 2017**), often in concert with each other (**Zhao et al., 2022**), however empirical studies demonstrating whether maturation of systemically derived monocytes into inflammatory synovial macrophages during OA is dependent on these TFs are needed.

To understand which stromal-derived signals recruit or activate immune cells, we extended the widely utilized cellular crosstalk toolbox CellChat to identify the ligand-receptor axes most perturbed by joint injury and most active in stromal-to-immune communication, in an unbiased fashion. By deriving probability ratios for all possible multi- and uni-directional stromal-immune signaling axes in healthy and injured joints, M-CSF signaling emerged as a major crosstalk axis activated by injury. M-CSF (and its alternative ligand, IL-34) has been shown to mediate monocyte-macrophage maturation via the M-CSF receptor in other disease contexts (**Lin et al., 2019**; **Aikawa et al., 2010**; **Preisser et al., 2014**), and monoclonal antibody-mediated M-CSF receptor inhibition ameliorated cytokine production in human RA synovial explants and mitigated disease severity in murine collagen-induced arthritis (**Garcia et al., 2016**). Our TF binding analysis revealed Pu.1 as a putative regulator of *Csf1r*, in agreement with previous reports (**Zhang et al., 1994**; **Aikawa et al., 2010**; **Krysinska et al., 2007**). Ongoing studies are seeking to understand whether synovial fibroblast-derived M-CSF activates putative regulators like Pu.1, to promote monocyte maturation toward a pro-inflammatory phenotype. Further work is also needed to describe which upstream pathways promote M-CSF and IL-34 ligand secretion by synovial fibroblasts, pericytes, and mast cells. Given the universal dependence of macrophages, both resident and recruited, on M-CSF signaling, dissecting the nuanced mechanistic distinctions such as ligand-receptor combination, ligand bioavailability, sending and receiving cell types, and spatial proximity, will be crucial if targeting M-CSF signaling is to be harnessed therapeutically in OA. In conjunction, accurately defining the temporal dynamics, ontogenies, and pathological vs protective functions of distinct macrophage subtypes in synovium will be integral if these cells are to be targeted for intra-articular treatment.

## Materials and methods

### Key resources table

| Reagent type (species) or resource | Designation | Source or reference | Identifiers | Additional information |
|---|---|---|---|---|
| Biological sample (*Mus musculus*) | C57Bl/6 mice | Jackson Laboratory | Strain #000664 | Male and female |
| Antibody | TruStain FcX PLUS | BioLegend | RRID:AB_2783137 | Clone S17011E (1:1000) |

*Continued on next page*

*Continued*

| Reagent type (species) or resource | Designation | Source or reference | Identifiers | Additional information |
|---|---|---|---|---|
| Antibody | Anti-mouse CD3-APC/Fire750 (rat monoclonal) | BioLegend | RRID:AB_2572117 | Clone 17A2 (1:100) |
| Antibody | Anti-mouse CD11b-BV605 (rat monoclonal) | BioLegend | RRID:AB_11126744 | Clone M170 (1:400) |
| Antibody | Anti-mouse CD11c-PE/Dazzle594 (Armenian hamster monoclonal) | BioLegend | RRID:AB_2563654 | Clone N418 (1:200) |
| Antibody | Anti-mouse CD19-FITC (rat monoclonal) | BioLegend | RRID:AB_2629813 | Clone 1D3/CD19 (1:100) |
| Antibody | Anti-mouse CD45-BV650 (rat monoclonal) | BioLegend | RRID:AB_2565884 | Clone 30-F11 (1:400) |
| Antibody | Anti-mouse F4/80-APC/R700 (rat monoclonal) | BD Horizon | RRID:AB_2869711 | Clone T45-2342 (1:400) |
| Antibody | Anti-mouse FceRIa-PE/Cy7 (Armenian hamster monoclonal) | BioLegend | RRID:AB_10640122 | Clone MAR-1 (1:100) |
| Antibody | Anti-mouse Ly6G-BV421 (rat monoclonal) | BioLegend | RRID:AB_10897944 | Clone 1A8 (1:200) |
| Antibody | Anti-mouse MHC Class II-BV421 (rat monoclonal) | BioLegend | RRID:AB_2650896 | Clone M5/114.15.2 (1:200) |
| Antibody | Anti-mouse NK1.1-BV510 (mouse monoclonal) | BioLegend | RRID:AB_2562216 | Clone PK136 (1:200) |
| Peptide, recombinant protein | Type IV Collagenase | Sigma | CAS 9001-12-1 | |
| Peptide, recombinant protein | Liberase | Sigma | 5401119001 | |
| Peptide, recombinant protein | DNaseI | Sigma | CAS 9003-98-9 | |
| Chemical compound, drug | TO-PRO-3 iodide | Invitrogen | T3605 | |
| Software, algorithm | FlowJo v10 | BD/Treestar | NA | |
| Software, algorithm | Seurat v4.1 | *Hao et al., 2021* | https://doi.org/10.1016/j.cell.2021.04.048 | |
| Software, algorithm | Cluster Identity PRedictor (CIPR) | *Ekiz et al., 2020* | https://doi.org/10.1186/s12859-020-3538-2 | |
| Software, algorithm | Monocle 3 | *Cao et al., 2019* | https://doi.org/10.1038/s41586-019-0969-x | |
| Software, algorithm | RCisTarget (SCENIC) | *Aibar et al., 2017* | https://doi.org/10.1038/nmeth.4463 | |
| Software, algorithm | CellChat | *Jin et al., 2021* | https://doi.org/10.1038/nmeth.4463 | |
| Other | scRNA-seq data of synovial cells from ACLR | NCBI GEO | GSE211584 | *Knights et al., 2023a* |
| Other | scRNA-seq data of synovial macrophages from RA | NCBI GEO | GSE134420 | *Culemann et al., 2019* |
| Other | scRNA-seq data of joint immune cells from ACLR | NCBI GEO | GSE200843 | *Sebastian et al., 2022* |
| Other | scRNA-seq data of hindpaw joint cells from arthritis | NCBI GEO | GSE184609 | *Muench et al., 2022* |

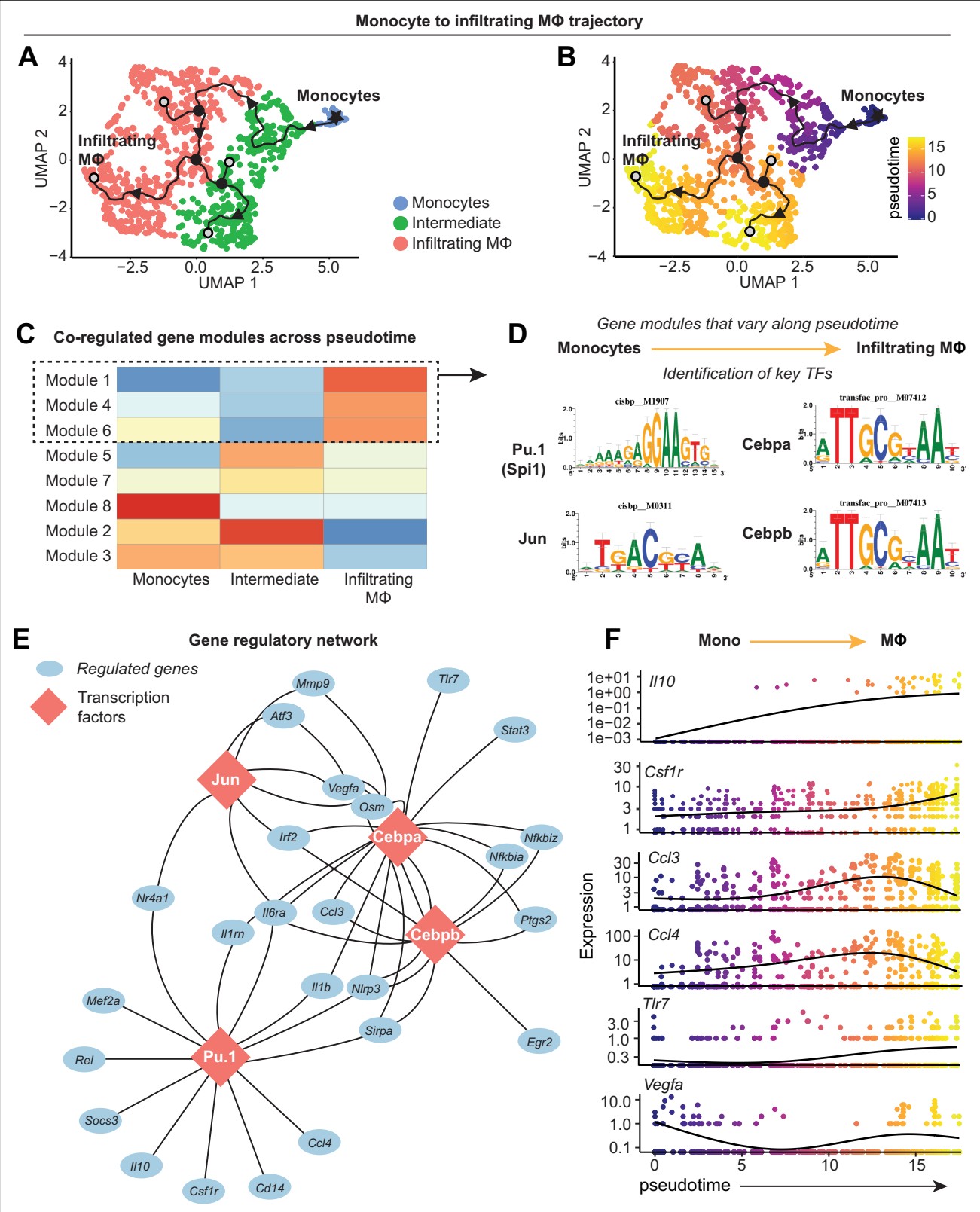

**Figure 6.** Transcriptional control of monocyte differentiation in synovium. (**A–B**) Pseudotime trajectory from monocytes to infiltrating MΦ showing directionality (arrowheads), starting point (root, star), branching points (nodes, black circles), and endpoints (termini, gray circles with black outline). Pseudotime scale is shown in (**B**). (**C**) Heatmap of gene module analysis of co-regulated genes across pseudotime trajectory from monocytes to infiltrating MΦ. Also see *Supplementary file 7*. (**D**) Modules 1, 4, and 6 were subjected to promoter screening for putative transcriptional regulators

*Figure 6 continued on next page*

*Figure 6 continued*

of module genes, using RcisTarget. Top directly annotated motif hits and their corresponding transcription factors (TFs) are shown. (**E**) Gene regulatory network for the TFs Pu.1 (Spi1), Jun, Cebpa, and Cebpb. (**F**) Pseudotime regression plots of selected gene from the gene regulatory network in (**E**). Monocytes to infiltrating MΦ, left to right.

The online version of this article includes the following figure supplement(s) for figure 6:

**Figure supplement 1.** Expression of key transcription factors across pseudotime from monocytes to macrophages.

| Reagent type (species) or resource | Designation | Source or reference | Identifiers | Additional information |
|---|---|---|---|---|
| Other | Bulk RNA-seq data of synovium | NCBI GEO | GSE271903 | *Bergman et al., 2024* |

## Mice

Male and female mice on a C57BL/6 background (Jax #000664) were used for all experiments. Mice were housed in ventilated cages of up to five animals, given chow food and water, on a 12 hr light/dark cycle. At time of injury, mice were 12–14 weeks of age, and were euthanized by $CO_2$ asphyxia. We induced PTOA using a previously reported and characterized model involving non-invasive ACLR (*Rzeczycki et al., 2021*). Sham mice were given anesthesia and analgesic, but not subjected to injury. All protocols were conducted in accordance with approved IACUC protocols at the University of Michigan.

## Perfusion of vasculature

To perfuse blood, mice were euthanized then washed with 70% ethanol. The thoracic cavity was opened through the diaphragm, and ribs were cut bilaterally to open the anterior ribcage, which was held with a hemostat. A butterfly needle was injected into the left ventricle and secured. A small incision was made in the right atrium to allow blood to escape, and the mouse was positioned to facilitate drainage from the thoracic cavity. The butterfly needle was connected to a perfusion pump filled with sterile PBS, which was perfused at a rate of 200 mL/hr for 15 min.

## Tissue harvest and digestion

Synovium was dissected from the medial, lateral, and anterior compartment of the knee, including the fat pad, as described previously (*Rzeczycki et al., 2021*; *Knights et al., 2023a*). Synovia were digested for 35 min in a 1.5 mL mixture containing DMEM with 400 µg/mL collagenase IV, 400 µg/mL liberase, 400 µg/mL DNaseI. 10 s of vortexing was performed at 0, 15, and 30 min.

Whole joints were isolated and digested in two stages, according to a detailed, published protocol from *Leale et al., 2022*. Hindlimbs from the mouse were removed by cutting the proximal leg muscles and dislocating at the hip. Muscles from around the knee joint were trimmed, taking care not to disturb the joint capsule, using a dissection microscope. PBS was used to keep the tissues moist throughout dissection. The knee joint was dislocated at the femoral and tibial growth plates to obtain just the whole joint, which was then transferred to a Petri dish for mincing with a scalpel. Minced joint tissue was subjected to 30 min of digestion, shaking at 90 rpm at 37°C. This first digestion, to yield the soft tissue fraction, used a 5 mL mixture comprised of 1% collagenase IV (wt/vol) and 400 µg/mL DNaseI in DMEM containing 5% fetal calf serum. After 30 min, the digestate was passed over a 70 µm strainer to collect undigested tissue. Undigested tissue was rinsed with DMEM and 5% fetal calf serum, then digested for a further 90 min at 90 rpm and 37°C. This second digestion, to yield the hard tissue fraction, used a 5 mL mixture comprised of 2% collagenase II (wt/vol) and 400 µg/mL DNaseI in DMEM with 5% fetal calf serum. At the conclusion of soft and hard tissue digests, single-cell suspensions were washed in DMEM containing 5% fetal calf serum then subjected to red blood cell lysis with cold ACK lysis buffer before proceeding to flow cytometry staining.

## Flow cytometry

After digestion of synovium or whole joints, single-cell suspensions were washed with FACS buffer (PBS containing 2% fetal calf serum and 1 mM EDTA) and 1 µL of FcX TruStain PLUS (BioLegend) was added to each sample. Cells were stained for 30 min at 4°C in the dark using fluorescently conjugated antibodies (*Supplementary file 1*). After staining, cells were washed with FACS buffer and passed

through 35 µm strainers into tubes for flow cytometry. TOPRO3 dye was added for determination of viability. An unstained control was included for all experiments, as well as single-stained control tubes and fluorescence-minus-one tubes for setting negative and positive populations for each color. Flow cytometry was performed on a BD LSRFortessa machine using FACSDiva software for data acquisition.

To visualize high-parameter flow cytometry data in two dimensions, dimensionality reduction was performed using the t-distributed stochastic neighbor embedding (t-SNE) algorithm in FlowJo v10 (*Belkina et al., 2019*). The automatic learning configuration was used (opt-SNE) and parameters were set to: iterations 1000; perplexity 30; learning rate 5498; KNN algorithm, Exact (vantage point tree); gradient algorithm, Barnes-Hut. All analysis was performed in FlowJo v10 (BD/TreeStar).

## Single-cell analysis of synovial immune cells

Data for the scRNA-seq analysis of synovial immune cells in the ACLR model of PTOA was drawn from GSE211584. Two biological replicates, each comprised of a male and a female synovium, were present for each of the following conditions: Sham (healthy, uninjured), 7 days ACLR, and 28 days ACLR. Quality control and filtering were performed using Seurat (R, v4.1.0) as described in the original publication (*Knights et al., 2023a*). As shown in *Figure 1—figure supplement 1A*, immune cell clusters (2, 5, and 8) were computationally subset from all synovial cells. Re-clustering and non-linear dimensionality reduction using uniform manifold approximation and projection (UMAP) was undertaken with dims = 1:35 and res = 0.2. Cluster identities were designated based on the expression of marker genes using the `FindAllMarkers` function and using the CIPR tool (*Ekiz et al., 2020*). Due to the low abundance of certain immune cell types and difficulty resolving between subsets of cell types, the `CellSelector` tool in Seurat was used to designate cluster identities in a supervised manner based on known marker genes.

## Recapitulation of publicly available datasets

We searched the NCBI Gene Expression Omnibus database using the search terms 'arthritis' and 'synovium'. Datasets from *Mus musculus* that did not exclude hematopoietic cells were retrieved and read into RStudio using Seurat. Quality control and filtering were undertaken faithful to the parameters and details provided in the originating publication methods. Where parameters were not provided, defaults were used: `min.cells` = 200, `min.features` = 200, `max.features` = 90th percentile cutoff, `mito.ratio` < 10%. Counts were normalized with the `NormalizeData` function using the `LogNormalize` method, then integrated using RPCA with `variable features` = 3000. Recapitulations for each dataset can be found in *Figure 2—figure supplement 1* (Culemann et al.), *Figure 2—figure supplement 2* (Sebastian et al.), and *Figure 2—figure supplement 3* (Muench et al.).

GSE134420, an scRNA-seq dataset produced by *Culemann et al., 2019*, was comprised of sorted CD45+ CD11b+ Ly6G- cells (macrophages/monocytes) from hindpaw synovia of mice subjected to the K/BxN serum transfer arthritis model of RA. No parameters were provided for quality control, so the default parameters above were used. For non-linear dimensionality reduction with UMAP, seven clusters emerged (dims = 1:30, res = 0.15). One cluster was identified as stromal cells (cluster 6), so was removed. Remaining clusters (immune only: 0, 1, 2, 3, 4, 5) were computationally subset and re-clustered using dims = 1:30 and res = 0.15, yielding five clusters, each with unique markers broadly corresponding to the original publication, although the Stmn1+ and Acp5+ macrophages fell into the same cluster.

GSE200843, an scRNA-seq dataset produced by *Sebastian et al., 2022*, was comprised of sorted CD45+ cells (all hematopoietic) from whole knee joints of mice subjected to the ACLR model of PTOA. Data were filtered using min.features>500 and min.cells=5, in accord with the original publication. We further excluded the top 10% of cells with the highest nFeatures using nFeatures<4500 and percent.mt<10, then normalized data using the `LogNormalize` method, and integrated data across all samples using reciprocal PCA, with variable features = 3000. Non-linear dimensionality reduction utilized dims = 1:30 and res = 0.04 to yield seven clusters. A stromal cluster (cluster 4) and an erythrocyte cluster (cluster 6) were detected in the ensuing object, based on marker gene expression. All other clusters (immune only: 0, 1, 2, 3, 5) were then computationally subset and re-clustered using dims = 1:25 and res = 0.07 to yield seven clusters.

GSE184609, an scRNA-seq dataset produced by *Muench et al., 2022*, was comprised of sorted live cells from hindpaw synovia of mice subjected to the GPI-induced model of RA. The authors

originally used dims = 1:20 for non-linear dimensionality reduction, and filtered based on nFeature and percent.mt cutoffs; however, the values for cutoffs and resolution were not provided. To recapitulate their data most faithfully, we used dims = 1:30 and res = 0.4, and obtained 21 clusters, representing all live sorted cells from the hindpaw synovium. Based on marker gene expression and cluster identity prediction by CIPR, we excluded non-immune cell clusters (1, 3, 10, 11, 12, 13, 14, 18, 19, 20) and computationally subset only the immune cells (0, 2, 4, 5, 6, 7, 8, 9, 15, 16,17). Immune cells were re-clustered using dims = 1:30 and res = 0.15, yielding eight clusters.

For detailed experimental methods, the reader is directed to the original cited publications for each dataset.

## Integration of datasets

The recapitulated immune objects for each dataset above were integrated using reciprocal PCA in Seurat with variable features = 3000. Non-linear dimensionality reduction was performed using dims = 1:30 and res = 0.2 to generate an integrated UMAP plot for all datasets. The resulting 10 clusters were annotated using `FindAllMarkers` and CIPR. Mast cells, due to their very low abundance, were clustered separately in a supervised manner using `CellSelector`, based on expression of marker genes *Il4*, *Kit*, *Fcer1a,* and *Mcpt4*.

## Analysis of macrophages in OA and RA datasets

To compare macrophages and monocytes between disease states, two clusters were computationally subset from the integrated object for all datasets (the macrophage and monocyte clusters). Non-linear dimensionality reduction was used to generate a UMAP with dims = 1:20 and res = 0.05, containing three clusters that were annotated using `FindAllMarkers` output. To assess DEGs in the macrophage cluster between RA (Culemann and Muench) and PTOA (Sebastian and Knights), `FindMarkers` was used. First, macrophages from control vs RA disease, and from control vs PTOA disease were compared to yield DEGs. Using a cutoff of padj<0.05, the DEGs from both control vs disease comparisons were then compared for overlapping genes and their directionality. To assess differentially regulated pathways in RA or PTOA macrophages, we used PantherDB (*Mi et al., 2021*) and the Gene Ontology database. First, DEGs unique to the RA vs control comparison, and unique to the PTOA vs control comparison, were separately entered into PantherDB for statistical overrepresentation analysis (Gene Ontology: Biological Pathways). To assess conserved pathways in RA and PTOA macrophages, we took DEGs common to both comparisons and with the same directionality and performed statistical overrepresentation testing using GO: Biological Pathways and GO: Molecular Function on PantherDB. All pathway analyses were performed using Fisher's exact testing, and false discovery rate was calculated to generate adjusted p-values (padj). Data were expressed as bubble plots using *ggplot2* v3.4.2 (*Hadley, 2016*).

## Macrophage clustering and trajectory analysis

Myeloid cell clusters from our scRNA-seq data were computationally subset from all immune cells in Seurat, then were exported into Monocle3 (*Cao et al., 2019*) for re-clustering to uncover subsets and perform trajectory analysis. Mast cells, DCs, and granulocytes were removed from the myeloid object, leaving only monocytes and macrophages, to mitigate the confounding variance from disparate cell types. The monocytes and macrophages were clustered with 15 dimensions, and `cluster_cells()` parameters of resolution = 2.5e-4 and k=11. For differentiation trajectory analysis, default `learn_graph` arguments were applied to generate a trajectory trail map, which robustly sequestered into two partitions. Differential gene expression analysis and functional pathway analysis were then performed on the three clusters confined to the first partition, containing cells with hallmarks of resident macrophages. Pairwise differential expression analyses were performed with the basal resident, resident-like A, and resident-like B macrophages clusters using `FindMarkers` in Seurat. To assess enriched pathways in each of these three clusters, DEGs from the prior pairwise analyses were used for statistical overrepresentation testing in PantherDB. Biological pathways enriched in any given cluster are indicated in figure panels, derived from genes that were upregulated in that particular cluster compared to the other. Pathway analyses were performed as described above in comparisons of PTOA and RA macrophages.

For more detailed analysis of how monocytes differentiate into infiltrating macrophages in synovium, we performed gene module analysis. The partition containing the monocytes and infiltrating macrophages was subset using `choose_graph_segments()` and re-clustered using 13 dimensions and a resolution of 1e-3 in `cluster_cells()`. An ncenter of 300 and minimal branch length of 15 were applied to generate trajectories. Gene modules, which are groups of co-regulated genes across a pseudotime trajectory, were identified using a resolution of 7e-3. Pseudotime regression plots of key genes were created using the `plot_genes_in_pseudotime` function.

### TF binding motif analysis

Genes from the three modules with high expression exclusively in the infiltrating macrophage cluster (the terminus of the monocyte-to-infiltrating macrophage trajectory) were submitted to RcisTarget (*Aibar et al., 2017*) for TF binding motif analysis. The mm9-tss-centered-5kb-7species.mc9nr.genes_vs_motifs.rankings.feather dataset was used to infer TFs responsible for upregulation of those genes. Significant, direct annotation-derived, high-confidence TF motifs (NES>3) expressed by at least 5% of macrophages and monocytes with a logcount>2 were studied. Select gene module genes with a maxRank of 2000 in these motifs were plotted into a network diagram using the visNetwork package.

### Intercellular communication analysis

CellChat (*Jin et al., 2021*) was used to infer ligand-receptor communications between stromal and immune cells clusters. In the `computeCommunProb()` function, the triMean-type analysis was applied. A minimum cell threshold of 10 was applied for each cluster and a significance threshold of 0.05 was applied when inferring communicating pathways. Line plots, heatmaps, and river plots were generated for each condition (Sham, 7 days ACLR, and 28 days ACLR). Significantly communicating pathways were ranked by ratio of crosstalk communication probabilities (stromal to immune or immune to stromal) relative to all communication probabilities to determine pathways most relevant to crosstalk. Highly ranked pathways were compared between Sham and ACLR conditions to identify crosstalk-relevant pathways most activated by injury. Hierarchy plots of significant communications were generated for CSF, which was identified as a top crosstalk pathway.

CellChat was used to infer outgoing signaling from macrophages, monocytes, and osteoclasts in OA (Knights, Sebastian) or RA (Meunch, Culemann). The triMean type was used in the `computeCommunProb()` function, with a minimal cell count of 10 per cluster. Significant pathways (p<0.05) were included for line plot, heatmap, and river plot generation. A default cutoff of 0.5 was used for river plots of significant outgoing communications.

## Acknowledgements

We would like to acknowledge the services offered by the University of Michigan Flow Cytometry Core. This work was supported by funding from the National Institutes of Health to AJK (K99AR081894) and TM (R01AR080035, R21AR076487) and from the Dr. Ralph and Marian Falk Medical Research Trust (Catalyst Award to TM). TM receives further unrelated research support from the National Institutes of Health (R21AR080502, R21AR082016, UC2AR082186) and the Department of Defense/CDMRP (GRANT13696744). AJK was supported by a Pioneer Postdoctoral Fellowship from the University of Michigan. ECF was supported by the Dr. Ralph and Marian Falk Medical Research Trust and National Institutes of Health (R01AR080035 to TM).

## Additional information

### Funding

| Funder | Grant reference number | Author |
| --- | --- | --- |
| National Institute of Arthritis and Musculoskeletal and Skin Diseases | R21AR076487 | Tristan Maerz |

| Funder | Grant reference number | Author |
|---|---|---|
| National Institute of Arthritis and Musculoskeletal and Skin Diseases | K99AR081894 | Alexander J Knights |
| Ralph and Marian Falk Medical Research Trust | Catalyst Award | Tristan Maerz |
| National Institute of Arthritis and Musculoskeletal and Skin Diseases | R01AR080035 | Tristan Maerz |
| National Institutes of Health | R21AR080502 | Tristan Maerz |
| National Institutes of Health | R21AR082016 | Tristan Maerz |
| Congressionally Directed Medical Research Programs | HT94252310327 | Tristan Maerz |

The funders had no role in study design, data collection and interpretation, or the decision to submit the work for publication.

#### Author contributions

Alexander J Knights, Conceptualization, Data curation, Formal analysis, Supervision, Investigation, Methodology, Writing – original draft, Writing – review and editing; Easton C Farrell, Data curation, Formal analysis, Methodology, Writing – original draft; Olivia M Ellis, Michelle J Song, Data curation, Formal analysis, Methodology; C Thomas Appleton, Conceptualization, Writing – original draft, Writing – review and editing; Tristan Maerz, Conceptualization, Formal analysis, Supervision, Funding acquisition, Writing – original draft, Project administration, Writing – review and editing

#### Author ORCIDs

Alexander J Knights ![ORCID] https://orcid.org/0000-0001-8402-2212
Tristan Maerz ![ORCID] https://orcid.org/0000-0002-4523-6012

#### Ethics

This study was performed in accordance with the highest ethical standards and with approval from the University of Michigan Institutional Animal Care and Use Committee, approval number PRO00009949. All injury procedures were performed under isoflurane-induced anesthesia, and an appropriate post-injury analgesic regimen was employed.

Reviewer #1 (Public review): https://doi.org/10.7554/eLife.93283.2.sa1
Reviewer #2 (Public review): https://doi.org/10.7554/eLife.93283.2.sa2
Author response https://doi.org/10.7554/eLife.93283.2.sa3

## Additional files

### Supplementary files

Supplementary file 1. Flow cytometry antibodies.

Supplementary file 2. Top 10 Cluster Identity PRedictor (CIPR) hits for each neutrophil cluster from the integrated object of all immune cells, all datasets.

Supplementary file 3. Ctrl vs post-traumatic osteoarthritis (PTOA) mac differentially expressed genes (DEGs).

Supplementary file 4. Ctrl vs rheumatoid arthritis (RA) mac differentially expressed genes (DEGs).

Supplementary file 5. Gene ontology outputs for post-traumatic osteoarthritis (PTOA) and rheumatoid arthritis (RA) macrophages.

Supplementary file 6. Gene ontology outputs for resident-like macrophages.

Supplementary file 7. Monocyte-to-infiltrating macrophage trajectory gene modules.
MDAR checklist

### Data availability

scRNAseq data have been deposited in GEO at GSE211584.

The following previously published datasets were used:

| Author(s) | Year | Dataset title | Dataset URL | Database and Identifier |
|---|---|---|---|---|
| Maerz T, Knights A, Farrell E | 2023 | Synovial fibroblasts assume distinct functional identities and secrete R-spondin 2 to drive osteoarthritis | https://www.ncbi.nlm.nih.gov/geo/query/acc.cgi?acc=GSE211584 | NCBI Gene Expression Omnibus, GSE211584 |
| Kroenke G, Culemann S, Klingberg A, Kirchner P, Ekici AB | 2019 | Locally renewing resident synovial macrophages provide a protective barrier for the joint | https://www.ncbi.nlm.nih.gov/geo/query/acc.cgi?acc=GSE134420 | NCBI Gene Expression Omnibus, GSE134420 |
| Sebastian A, Loots GG | 2022 | Single-cell RNA-Seq reveals changes in immune landscape in post-traumatic osteoarthritis | https://www.ncbi.nlm.nih.gov/geo/query/acc.cgi?acc=GSE200843 | NCBI Gene Expression Omnibus, GSE200843 |
| Muench DE, Sun Z | 2022 | A pathogenic Th17/CD38+ macrophage feedback loop drives inflammatory arthritis through TNFa | https://www.ncbi.nlm.nih.gov/geo/query/acc.cgi?acc=GSE184609 | NCBI Gene Expression Omnibus, GSE184609 |
| Bergman RF, Lammlin L, Junginger L, Farrell E, Goldman S, Darcy R, Rasner C, Obeidat AM, Malfait A, Miller RE, Maerz T | 2024 | Sexual dimorphism of the synovial transcriptome underpins greater PTOA disease severity in male mice following joint injury | https://www.ncbi.nlm.nih.gov/geo/query/acc.cgi?acc=GSE271903 | NCBI Gene Expression Omnibus, GSE271903 |

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
