## [Editor Report · eLife assessment]

This study provides **useful** information by identifying the cell type (macrophages) in synovial tissues involved in the pathogenesis of post-traumatic osteoarthritis (OA) and clarifying distinct transcriptomic signatures that may be a good therapeutic target for OA. However, the analysis performed so far is **incomplete**, with a main weakness being the lack of data to confirm the authors' speculation about the underlying mechanisms.

---

## [Referee Report · Reviewer #1 (Public review)]

Summary:

The authors recently reported a scRNA-seq-based study focused on synovial fibroblasts using a mouse model of post-traumatic OA (Ref. 21). In the present manuscript, they reanalyzed the scRNA-seq data to investigate the diversity and roles of macrophages. In addition to their original scRNA-seq data (Ref. 21), they utilized the deposited data of other OA or RA models (Ref. 25-27) and compared cell types in the synovium. The authors extracted the macrophage/monocyte group, compared differentially expressed genes (DEGs) between OA and RA synovium, and analyzed macrophage subsets, including trajectory analysis. They further estimated the crosstalk between stromal and immune cells via M-CSF signaling, and transcription factors for monocyte differentiation.

Strengths:

The descriptions are comprehensive, based on the scRNA-seq data including the original and other independent studies.

Weaknesses:

Meanwhile, methods of sample preparation must be different, for example, the extent and location of excised synovium. The comparison with other studies is meaningful and informative; however, caution should be exercised regarding the potentially significant impact of methodological differences on the analysis results.

The various data obtained from these technologies are comprehensive and useful; however, they are just estimates. Without confirmation by experiments, it is impossible to determine how much of it can be believed. This issue is not limited to this paper.

Most of all signaling pathways and molecules described in the latter part of this study are previously known.

---

## [Referee Report · Reviewer #2 (Public review)]

Summary:

The manuscript by Knights et al set out to identify the specific immune cells and their contribution to the development of osteoarthritis. They performed a comprehensive analysis of scRNA-seq and flow cytometry using different stages of the PTOA model and sought to identify specific synovial macrophages in OA. Computational analysis revealed that M-CSF signaling in synovium plays an important role in stromal-immune crosstalk in OA. They also found that four transcription factors including Pu.1, Cebp-alpha, Cebp-beta, and Jun regulate the differentiation of monocytes into pro-inflammatory synovial macrophages in OA.

Strengths:

The main strength of this study is the profiling of immune cells which will be a valuable resource for better understanding the pathogenesis of OA. The work is technically sound, and the level of analysis of gene expression, clustering, cell-cell communication, and dynamic changes in gene modules over time is state-of-the-art.

The reviewer appreciates that the authors uncovered the transcriptional network that regulates the differentiation of synovial macrophages in OA. In addition, the identification of M-CSF signaling as a major crosstalk axis in OA development is also intriguing.

Weaknesses:

Although the scRNA-seq analysis of immune cells in OA is quite convincing, the data has been rather descriptive and superficial at this stage. The authors did not show the in vivo significance of their findings in OA development.

---

## [Author Response]

We greatly appreciate the feedback provided by reviewers on this manuscript. One of our key objectives was to provide a comprehensive, detailed resource for researchers using single-cell transcriptomics to study arthritis, especially immune cells like macrophages. We strived to perform thorough, wide-ranging analyses that are both accessible and useful to other scientists in the field, and that we hope will serve as the basis for many future avenues of study. As such, we acknowledge that this work is a “first step”, providing a strong descriptive foundation with some mechanistic insight that we and others will continue pursuing. Preliminary studies in our laboratory seeking to dissect signaling mechanisms associated with the M-CSF pathway have illuminated how complex and context-dependent this signaling is, which is an important consideration for future in vivo investigations. Further, it is indeed true that attempting to harmonize transcriptomic data across studies, models, laboratories, and dissection/processing methods is fraught with difficulty and prone to misinterpretation – and we made an effort to highlight this in our manuscript, particularly with respect to where synovial immune cells were recovered from, and how. We encourage healthy discussion within the field for developing shared, unified protocols for harvests and processing upstream of transcriptomic experiments.